# Drivers of Shrub Community Assembly in Semi-Arid Ecosystems: Integrated Evidence from Environmental Stress on the Western Loess Plateau

**DOI:** 10.3390/biology14111465

**Published:** 2025-10-22

**Authors:** Minghao Li, Han Dang, Jiawei Du, Dan Liu, Tong Yu, Jinshi Xu, Biao Han, Ping Ding, Dechang Hu

**Affiliations:** 1School of Life Sciences, Ludong University, Yantai 264025, China; lmh17658123505@163.com; 2China ENFI Engineering Co., Ltd., Beijing 100038, China; danghan_1102@163.com; 3School of Life Sciences, Northwest University, Xi’an 710069, China; 4Wendeng Management Bureau of Kunyushan National Nature Reserve, Weihai 264421, China; djw1341112141@163.com (J.D.); 18769116712@163.com (T.Y.); 5Shandong Provincial Center of Forest and Grass Germplasm Resources, Jinan 250100, China; 1821618@163.com (D.L.); hanbiaook831228@163.com (B.H.); 18953130256@wo.cn (P.D.)

**Keywords:** environmental stress, shrubland, semi-arid area, biodiversity, phylogenetic structure

## Abstract

Shrubland is the main vegetation on the western Loess Plateau, which has harsh environment conditions. Under stress environments, such as low temperature, human disturbance, and drought, the diversity levels and coexistence mechanisms of different types of shrub communities are different. It is necessary to clarify how the diversity and community assembly processes of the shrublands change in this area along these stress habitat gradients in order to understand their response mechanisms to stress conditions. Low temperature is a limiting factor affecting the diversity of shrub communities in this area. Different shrubland types are all driven by deterministic community assembly processes, even though the environmental driving forces may vary. This study provides insights into the vegetation restoration and stability processes in fragile habitat areas.

## 1. Introduction

As one of the most severely degraded regions, more than 60% of the Loess Plateau is affected by soil erosion [1,2], which leads to severe environmental stress. Habitat stress is particularly severe in the western part of the Loess Plateau due to severe wind erosion and water erosion, as well as long-term human interference [3]. Shrubland is the main vegetation type in this area, which is of great significance to soil and water conservation and ecosystem stability in this area [4]. Therefore, conducting a comparative study on the existing shrub communities in the western the Loess Plateau can elucidate the causes of diversity differences among various types of shrublands. This study may also uncover the factors driving the development of distinct shrub communities, thereby offering theoretical support for vegetation restoration and biodiversity conservation in the ecologically sensitive western Loess Plateau.

The State Forestry Administration of China has designated “shrub forest land as specially stipulated by the state”, which specifically refers to desert shrublands distributed in extremely arid, arid, and semi-arid areas with an average annual precipitation of less than 400 mm. Alpine shrublands occur above the upper limit of tree forest distribution (vertical distribution). Secondary shrublands emerged throughout the succession process as a result of intensive, large-scale human disturbances, exemplified by areas logged and later allowed to regrow, or by cropland converted back to forest. Typical desert, alpine, and secondary shrub communities are distributed throughout the western Loess Plateau. Previous studies have shown that environmental stress factors affect vegetation growth, influencing diversity structure [5]. The ecological processes responsible for forming the diversity levels are clearly distinct. Furthermore, diversity can be used as an indicator to characterize the relationship between plants and environmental factors [6], which can explain how plants respond to stress habitats. Many scholars have previously studied the environmental factors affecting shrub growth in this region. One study found that water is the main factor [7], while other research found that soil nutrients can influence shrub growth [8]. However, there is little research on the primary environmental factors that restrict the growth of different types of shrubs and the assembly of communities. Therefore, it is necessary to explore the ecological processes that affect the formation and maintenance of diversity and to clarify the relative importance of specific environmental factors in the assembly of different types of shrub communities in the western Loess Plateau.

Phylogeny refers to the phylogenetic relationships between species within a community. In the absence of functional trait data, the phylogenetic relationship of a community can be regarded as a “proxy” for the functional traits of species within the community. It is assumed that a closer phylogenetic relationship among species in a community indicates that they have more similar functional traits and are therefore easier to filter by the same environment [9]. Species with distant phylogenetic relationships exhibit significant differences in functional traits, and niche differentiation is sufficient to avoid similarity limitations [10]. Therefore, using phylogenetic methods to study diversity and community assembly mechanisms has advantages such as simplicity, efficiency, and significant ecological significance. Meanwhile, species diversity can intuitively reflect the taxonomic differences among coexisting species within a community, thereby revealing the spatiotemporal variation patterns of community species composition [11]. Therefore, considering both species diversity and phylogenetic diversity can help clarify the process of community assembly and diversity maintenance mechanisms [12].

Plant diversity and community assembly are influenced by climate, terrain, and soil nutrients [13,14,15]. Moreover, the differences in species fitness caused by habitat filtering and similarity limitations can lead to the formation of different types of functional communities in different environments [16]. These deterministic processes in the community assembly comprise the main theoretical foundation of the niche hypothesis. Therefore, exploring the patterns of change in plant community diversity and community assembly along gradients of climate factors, soil factors, and terrain factors can clarify the main driving forces and ecological processes in community formation, which can help explain the relative importance of these factors and processes in maintaining diversity.

This study focused on the three types of shrublands in Shaanxi, Gansu, and Ningxia in the western Loess Plateau of China. Based on the literature records of vegetation classification and habitat types, we distinguished desert shrublands, alpine shrublands, and secondary shrublands, and then conducted comparative studies along some environmental stress factors. The species diversity, phylogenetic diversity, and community assembly patterns of the three types of shrublands are also discussed. Through the above research, we attempt to answer the following questions: (1) Which environmental factors dominate the formation of shrub community diversity in semi-arid areas of the Loess Plateau? (2) What ecological processes drive the community assembly of different types of shrublands in semi-arid regions of the Loess Plateau? Regarding the above research questions, we hypothesized that (H1) mean annual temperature (MAT) is the primary driver of diversity across all shrubland types, and (H2) alpine and desert shrublands are primarily assembled by habitat filtering, while secondary shrublands show signs of competitive exclusion due to different disturbance histories.

## 2. Materials and Methods

### 2.1. Study Area

This study was conducted across the provinces of Gansu, Shaanxi, and Ningxia in Northwestern China. The study area encompasses numerous distinct geographical units, presenting complex and varied topography and diverse landform characteristics. The elevation of the study area spans from 550 to 3700 m, the mean annual temperature ranges from −5 to 17 °C, and the mean annual precipitation ranges from 130 to 1000 mm. We selected a total of 49 shrublands, including 19 desert shrublands, 12 alpine shrublands, and 18 secondary shrublands. The sampling locations for each type of shrubland were selected from the most typical areas of that type for investigation. The specific shrubland types and research sites are shown in Figure 1.

### 2.2. Investigation Methods

Field investigations were conducted during the plant growth period from July to August. Among the 49 sample sites mentioned above, three quadrats were selected from each site location, including those with typical vegetation (verified by the recorded plant species and local vegegraphy), mild disturbance (no obvious signs of human activity), and close proximity. The shrubland distribution area where the quadrats were located was not less than 25 m × 25 m. Each quadrat had an area of 5 m × 5 m, and the distribution radius of the three quadrats was kept within 250m. For each quadrat, we captured its coordinates (latitude and longitude), elevation, gradient, and position on the slope. We recorded the species name, abundance, and coverage of each plant species in the quadrats. When conducting further analysis, the species lists of the three quadrats were pooled into a single composite plot. Each sample site covered an actual survey area of 75 m^2^. A total of 191 shrub species were recorded across all survey sites.

### 2.3. Environmental Data

In addition to the geographical factors recorded in the field surveys, soil samples were also collected. First, litter on the soil surface was cleared. Following the project’s technical protocol, three soil sampling points were established along the diagonals within each 5 × 5 m quadrat. Soil samples were taken from a depth of 0–20 cm at each point. Samples obtained from the three sampling points within each 5 × 5 m quadrat were mixed and placed into self-sealing bags. Three composite soil samples were obtained from each large site and transported to the laboratory for physicochemical analysis.

The soil samples were passed through an 80-mesh sieve, and then identical sample masses were used for testing. Soil pH was measured with a pH meter, TOC was measured using the oil bath titration method with sulfuric acid–potassium dichromate, TN was measured using a Kjeldahl nitrogen analyzer, and TP was measured with a spectrophotometer using standard methods. We took the averages as the soil property data for the sample site.

According to the latitudes and longitudes of the sample sites, we extracted climate factors from a 1 × 1 km grid map in the WoldClim database (http://www.worldclim.org/). Based on previous research and pre-analysis, this study selected the MAT as a climate factor for analysis [17], and the data was obtained through the Raster package of R 4.1.2 software.

### 2.4. Data Analysis

To explore whether the habitat environments of similar shrub communities exhibited similarities, we used principal component analysis (PCA) to assess environmental differences among various types of shrublands and performed PCA mapping in Canoco 4.5 software. The environmental factors involved in PCA are shown in Table A1.

To investigate whether there are differences in the diversity patterns of different types of shrub communities, we selected species richness as an indicator of species diversity [18], and Faith’s PD (the total length of evolutionary branches of all species within the community) as an indicator of phylogenetic diversity [19]. Before conducting phylogenetic analysis, it was necessary to construct a suitable phylogenetic tree in order to obtain accurate conclusions. In this study, to avoid false positive results caused by a larger species pool, species from desert shrublands, alpine shrublands, and secondary shrublands plots were selected to construct total species pools, which were used as background data to construct phylogenetic trees for the three different types of communities. The phylogenetic tree used in this study was derived from the time-calibrated ML tree of Zanne et al. (2014), which is fully bifurcating but does not explicitly address whether polytomies were resolved or retained [20]. Therefore, no polytomies were further analyzed or corrected in our analysis progress. The construction of phylogenetic tree used phylocomr package of R 4.1.2. Faith’s PD was calculated using the phylo package of R 4.1.2 [21].

To clarify the causes underlying the formation of diversity in different types of shrub communities, this study selected the nearest taxon index (NTI) as an indicator of phylogenetic structure. The null model used for NTI calculation was taxa.labels, which randomizes species labels across the phylogeny while keeping species occurrence frequencies and plot richness constant. This model tests whether the observed phylogenetic structure deviates from random expectations in phylogeny distribution. The specific calculation method for NTI is as follows:$$NTI=-1\times \frac{{MNTD}_{observed}-{MNTD}_{randomized}}{{sdMNTD}_{randomized}}$$

In the formula, MNTD_observed_ is the observed value of MNTD (mean nearest taxon distance), and MNTD_randomized_ is the MNTD value randomly generated by the null model. *sd* represents the standard deviation. The null model was automatically generated by the software and calculated 999 times. NTI > 0 indicates phylogenetic aggregation; NTI < 0 indicates phylogenetic divergence; NTI = 0 indicates phylogenetic randomness [9]. The aggregation and divergence of phylogenetics can explain the key processes that affect community assembly, thereby elucidating the maintenance mechanism of community diversity. The calculation of NTI was carried out in the COMSTRUCT module of Phylocom 3.0 software.

To explore the impact patterns of environmental factors on the diversity and community assembly of different communities. We conducted nonlinear modeling in the preliminary processing stage. However, the goodness-of-fit was not satisfactory. Therefore, we ultimately selected the linear model, which demonstrated superior performance. General linear analyses were conducted on the relationships between various environmental factors and diversity, phylogenetic structure, and diversity indicators. This study analyzed the linear relationship between MAT, soil pH, TOC, TN, TP, slope, and diversity/phylogenetic structure data to explore the impact patterns of environmental factors on different shrubland diversity patterns. Linear fitting and plotting were completed in Origin 8.5 Pro.

To clarify the main environmental factors that affect the assembly of shrub communities, a variance decomposition-based relative contribution analysis was conducted on environmental factors using the vegan package in R 4.1.2. The dependent variable for variance decomposition is MNTD_observed._

## 3. Results

### 3.1. Changes in Diversity of Different Types of Shrublands Across Environmental Stress Gradients

In this study, climate factors, soil factors, and terrain factors of different types of shrublands were all included in the PCA. The results indicated that PC1 and PC2 could explain 92.4% of all environmental factor changes (with PC1 explaining 73.7%; Figure 2). In the PCA, deserts, alpine, and secondary shrublands showed relatively clear distribution ranges and boundaries. The PC1 axis was mainly related to MAT, soil pH, slope, etc., while the PC2 axis was mainly related to MAT, TOC, TP, and TN (Table A1).

We conducted regression analysis on the species diversity and phylogenetic diversity among different shrubland types along gradients of climate factors, soil factors, and terrain factors. The results indicated that the species diversity of alpine shrublands increased along the mean annual temperature, while the species diversity of desert shrublands decreased along the mean annual temperature. However, there was no significant correlation between species diversity of secondary shrublands and mean annual temperature (Figure 3a). The variation patterns in phylogenetics diversity of different types of shrublands along the MAT gradient were consistent with species diversity (Figure 3b). In addition, the mean annual temperature range of different types of shrublands had clear boundaries.

Both the species diversity and phylogenetic diversity of desert shrublands decreased with an increasing pH gradient. Both the species diversity and phylogenetic diversity of alpine shrublands and secondary shrublands showed no significant correlation with pH (Figure 4a,b). With increasing TOC, the species diversity and phylogenetic diversity of secondary and desert shrublands also increased. However, the species diversity and phylogenetic diversity of alpine shrublands were not significantly correlated with TOC (Figure 5a,b). With increasing of TP, the species diversity and phylogenetic diversity of desert shrublands increased (Figure 6a,b). With increasing TN, both the species diversity and phylogenetic diversity of desert shrublands also increased (Figure 7a,b).

Slope was positively correlated with phylogenetic diversity. We further examined the variation patterns of species diversity and phylogenetic diversity of different types of shrublands along the slope gradient. The results showed that the species diversity and phylogenetic diversity of desert shrublands increased with increasing slope, while the species diversity of secondary shrublands decreased with increasing slope (Figure 8a,b).

### 3.2. Relative Importance of Various Environmental Factors Affecting Shrubland Phylogenetic Relationships

To further clarify the contribution of various environmental stress factors to the diversity and formation mechanism of shrublands, we conducted variance decomposition on the relative contribution of climate factors, soil factors, and terrain factors to MNTD. The results showed that environmental factors explained 25% of the variance in shrubland phylogenetic relationships. Among them, the climate factor (MAT) independently explained 14% of the variation in shrubland phylogenetic relationships. The soil factors (pH, TOC, TP, and TN) independently explained 4%. The combined effect of the terrain factor (slope) and climate factor (MAT) explained 2%, while the combined effect of the terrain factor (slope) and soil factors (pH, TOC, TP, and TN) explained 5%. In addition, 75% of shrubland phylogenetic relationships could not be explained by three types of environmental factors (Figure 9).

### 3.3. Phylogenetic Structure of Different Types of Shrublands Along Environmental Gradients

To further determine which ecological process dominates the assembly of shrub communities, phylogenetic structure analysis was conducted on the gradients of various environmental factors among different shrubland types. The results showed that there was no significant correlation between MAT and NTI in any of the shrub communities. The NTI of secondary shrublands decreased along the MAT gradient; however, there was no significant correlation between the NTI and MAT in desert shrublands or alpine shrublands (Figure 10a). The terrain factor (slope) was negatively correlated with the NTI. The NTI of alpine shrublands decreased with increasing slope, while the NTI of secondary shrublands and desert shrublands showed no significant correlation with slope (Figure 10b).

## 4. Discussion

### 4.1. Mean Annual Temperature Shapes the Diversity Patterns and Assembly Processes of Three Types of Shrub Communities

The level of diversity within a community is an important indicator of the community assembly process, which can reflect the ecological characteristics and formation mechanisms of the community in multiple dimensions. Previous studies have found that the diversity level of a community is related to its habitat. Environmental factors such as climate, terrain, and soil may all affect the diversity level of a community [22,23,24]. In addition, the patterns of species, phylogenetic, and functional diversity may be similar or different across different environmental gradients [25]. Our results also showed that in the western Loess Plateau, the patterns of species diversity and phylogenetic diversity along environmental gradients were consistent, which is consistent with previous research in this area [26]. We found that different types of shrublands showed relatively clear distribution ranges and boundaries, indicating that there are obvious environmental differences among the different types of shrublands. This indicates that these selected environmental factors may be the causes of colonization and growth of different types of shrublands. It is important to note that soil factors and terrain factors could only affect the diversity of no more than two types of shrublands simultaneously. Furthermore, there was no clear boundary between the three types of shrublands regarding these environmental factors. In contrast, the linear relationship between diversity and MAT also indicates that there were clear boundaries between different types of shrub communities along the mean annual temperature gradient. Therefore, the mean annual temperature may be the main cause of changes in the diversity of most shrublands in the region.

We further analyzed the impact of various environmental factors on the diversity patterns of different types of shrub communities. We found that there was a significant positive linear correlation between the community diversity level of alpine shrublands and the mean annual temperature, which is consistent with previous studies [27]. We speculate that this may be due to lower temperatures in high-altitude areas, where mean annual temperature is a limiting factor that causes environmental filtering and exclusion of most species. Furthermore, this led to diversity variation in the temperature gradient [28]. The diversity level of desert shrubland communities was significantly negatively correlated with mean annual temperature. We hypothesize that this may be due to the intensified evaporation caused by high temperatures. Owing to the rapid loss of soil available water, only deep-rooted drought tolerant species can survive, resulting in community homogenization and loss of diversity [29]. There was no significant correlation between the diversity level of secondary shrubland communities and mean annual temperature. We speculate that this may be because the distribution area of such shrublands in this region has been severely disturbed by human activities. Additionally, this has affected the trend of diversity along other gradients, thereby masking the influence of temperature on the diversity of secondary shrublands.

Our research also found that environmental factors such as soil and terrain can affect the diversity level of a certain type of shrubland community. For example, the diversity level of desert shrublands decreased with increasing pH. Previous studies have shown that the soil in desert areas tends to be alkaline in the upper layer due to weak leaching processes [30]. The growth of plants requires a suitable soil pH environment [31], but the alkaline environment in desert areas is no longer suitable for the growth of shrubs, and the strong habitat filtration process has led to the loss of diversity. Our research also found that the community diversity of desert shrublands and secondary shrublands was positively correlated with TOC. It is worth noting that the TOC of both desert shrublands and secondary shrublands was at a relatively low level. For desert shrubland distribution areas, the soil formation process of sandy soil lacks the participation of humus. For the distribution area of secondary shrublands, this may be due to severe human interference in the early stage of succession, as most of the organic matter in the soil has been destroyed. Therefore, although the diversity of some shrublands along the TOC gradient showed a changing trend, this trend may not be universal. In addition, the diversity level of desert shrub communities was positively correlated with TP and TN. This may be due to the poor texture of desert soil, where soil nutrients become factors that restrict the diversity level of the community. After improving the soil nutrient conditions, the diversity of shrub communities in the desert area will show an increasing trend. In terms of terrain factors, the diversity level of desert shrublands was positively correlated with slope. The phylogenetic diversity of all the three types of shrubs also decreased with increasing slope. However, the slope of the sampling points in the desert area is generally low. The reason for this trend may be that the areas with larger slopes are not covered by sandy soil and instead have better site conditions. The diversity level of secondary shrublands was negatively correlated with slope, which is consistent with previous research results. This is because areas with steeper slopes have more severe soil erosion and poor soil moisture and nutrients, leading to the loss of diversity [32].

Additionally, we found that the mean annual temperature was the main factor affecting the phylogenetic structure within shrub communities. Previous studies have also found that temperature is the main factor affecting the shrub community assembly, especially in high-altitude areas with lower temperatures [33,34]. In addition, soil factors such as pH, TOC, TP, and TN, as well as terrain factors such as slope, can also affect shrub community assembly in this area to some extent [35,36], but they are not the key factors.

Our research had certain limitations. The three quadrats we established served as technical replicates of the local community. The species lists from these three quadrats were pooled into a single “plot” for analysis. This masked the spatial autocorrelation caused by environmental conditions and dispersal limitations. The reason for this design is that the study relied on a survey project with a standard quadrat size of 5 m × 5 m. However, for the alpine and desert shrublands in the western Loess Plateau, a 5 m × 5 m quadrat size was highly likely to capture only a single species, making it impossible to calculate the community assembly pattern. Therefore, we established three 5 m × 5m quadrats within the shrublands at a local scale (>25 m × 25m), attempting to obtain the same community’s symbiotic species in different microhabitats whenever possible. In our future work, we will establish multiple truly independent replicate sites for each shrubland, spaced far enough apart to ensure spatial independence.

### 4.2. Deterministic Processes Drive Community Assembly of Three Types of Shrublands on the Loess Plateau

A central goal of community ecology is to understand how diversity within a community is formed [37]. Niche theory posits that the drivers shaping community species composition form two opposing processes: habitat filtering and similarity limitations [9]. Neutral theory posits that species diversity within a community arises from dispersal limitation and ecological random drift [38]. Recent studies have found that environmental filtering, a deterministic process, dominates community assembly [39]. In certain environments, the deterministic process of competitive exclusion plays a major role in community assembly [40]. Interestingly, in recent years, studies have shown that the patterns of certain plant communities are influenced by both deterministic and stochastic processes [41]. We found that along the mean annual temperature gradient, the phylogenetic structures of alpine shrublands, desert shrublands, or secondary shrublands were mostly clustered (39 out of 49 samples had NTI > 0). This indicates that the shrub communities in the western part of the Loess Plateau are mostly affected by the environmental filtration process [33]. Competitive exclusion also has a certain impact on the community assembly of shrublands (a small number of sample points had NTI < 0). Of course, the assembly of shrub communities may also be influenced by stochastic processes, such as dispersal limitation. In addition, there was a significant negative correlation between the NTI of secondary shrublands and mean annual temperature, that is, with the mean annual temperature increasing, the phylogenetic structure showed a clear trend from aggregation to divergence. This may be due to the weakening of environmental filtering intensity following the rise in mean annual temperature. Meanwhile, due to the intense human activities in the distribution area of secondary shrublands, species from other regions may have been introduced, enhancing the interactions between organisms. This combined effect may lead to a pattern of phylogenetic from convergence to randomness and then to divergence. Due to poor water and heat conditions and harsh environmental filtering, desert and alpine shrublands are able to establish similar phylogenetic relationships among shrub types in the region [42,43], thus exhibiting a trend of phylogenetic convergence. At the same time, the mean annual temperature of desert and alpine shrublands is generally lower. Expanding a limited temperature gradient cannot eliminate environmental filtration caused by insufficient hydrothermal conditions. Therefore, the phylogenetic structure did not undergo significant changes with temperature.

Another environmental factor, slope, was found to have a significant negative correlation with the NTI. Phylogenetic structure tends to diverge from aggregation, with this trend being more pronounced in alpine shrublands. Previous studies on tree forests have generally shown that slope affects the assembly process of tree communities [44]. The greater the slope, the worse the site conditions, and the more concentrated the community phylogenetic relationships, which is contrary to our research conclusion [45]. Perhaps the steep terrain increases the heterogeneity of microhabitats, providing more ecological niches, weakening the filtering effect of the environment, and allowing distant shrub species to coexist. In addition, the trend of changes in the phylogenetic structure along the slope gradient may also be caused by changes in human disturbance, which has a greater impact on the community than other environmental factors [46]. Generally speaking, areas with gentler slopes experience more intense human disturbance, while steeper slopes see reduced human disturbance. This attenuation of environmental filtering through human disturbance allows distantly related species to coexist. Some scholars also believe that due to dispersal limitations, competitive exclusion may be more important for lightly disturbed and undisturbed communities [40]. The phylogenetic changes in alpine shrubs along the slope gradient may also be caused by intensified competitive exclusion due to dispersal limitations. By reviewing the above phenomenon, we believe both environmental filtering and similarity limitation processes have played important roles in the community assembly of shrublands.

## 5. Conclusions

This study examined the diversity and phylogenetic structure of deserts, alpine, and secondary shrublands in the western Loess Plateau, which experiences severe environmental stresses. We explored the environmental causes of diversity formation among different types of shrublands in this region based on the ecological processes of community assembly. We found that environmental stress factors influencing the assembly of different types of shrub communities exhibit significant differences. Among them, the mean annual temperature affects the colonization of different types of shrublands. The results of variance decomposition also indicate that temperature may be the key factor leading to changes in community diversity and assembly patterns. A notable finding is that steeper slopes in alpine shrublands exhibit increasingly divergent community phylogenetic relationships. This may be caused by microhabitat heterogeneity and reduced human disturbance on steep slopes. In the western Loess Plateau, the community assembly of different types of shrublands has been influenced by deterministic processes, such as environmental filtering and similarity limitations, supporting the niche hypothesis. This study also had certain limitations. The selection of sample areas was influenced by project standards, which may have an impact on the research results. The research findings are generally constructive for the conservation of shrubland vegetation. In the ecologically fragile areas of the western Loess Plateau, the conservation of shrub vegetation should account for the impacts of environmental factors such as human disturbance and annual average temperature.

## Figures and Tables

**Figure 1 biology-14-01465-f001:**
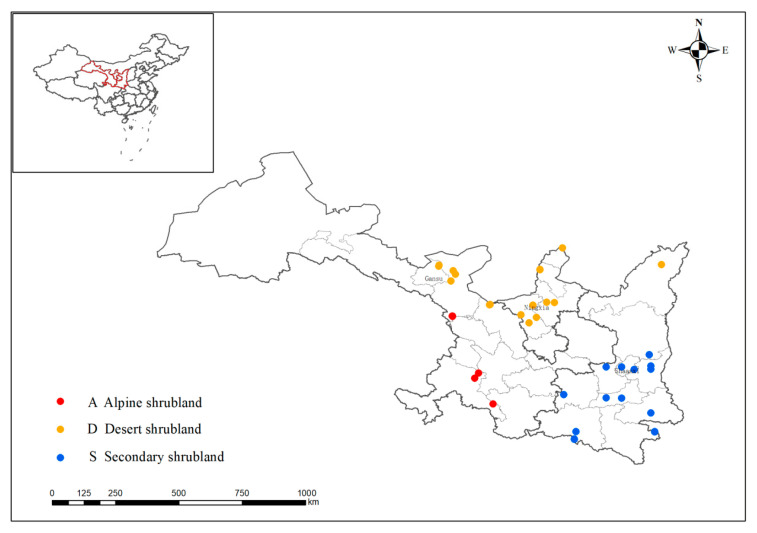
Study site map.

**Figure 2 biology-14-01465-f002:**
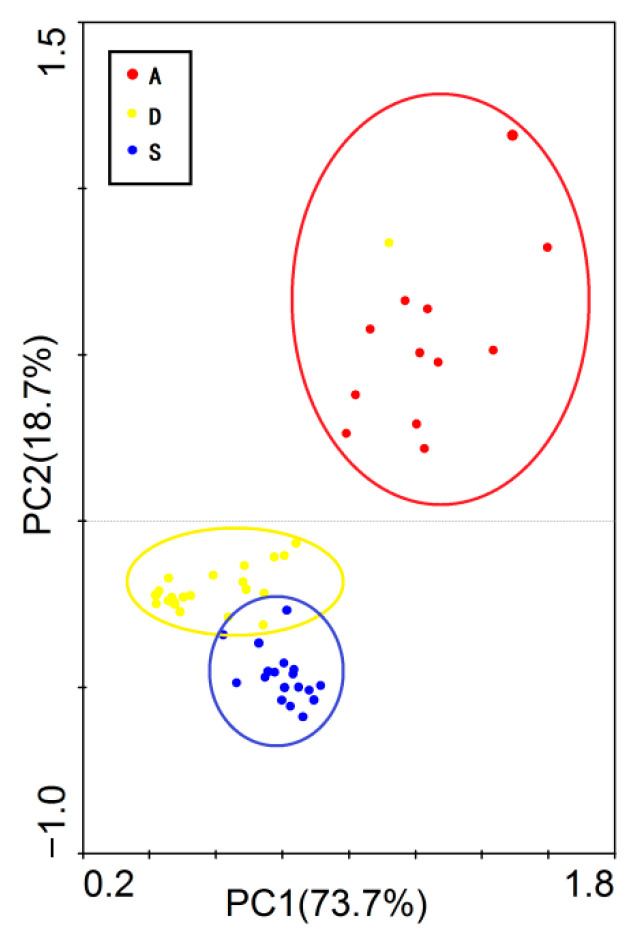
Principal component analysis of environmental factors in different shrubland types. A refers to alpine shrublands; D refers to desert shrublands; S refers to secondary shrublands.

**Figure 3 biology-14-01465-f003:**
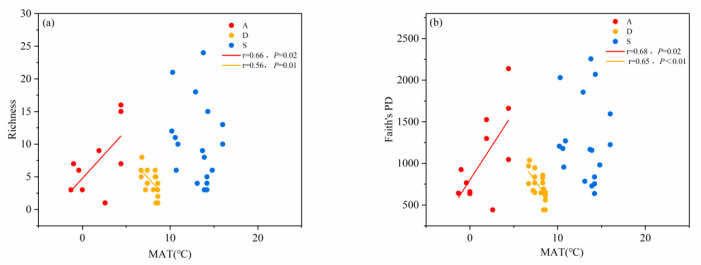
(**a**) Changes in species richness and (**b**) changes in phylogenetic diversity of different shrubland types along the mean annual temperature gradient. The red dots represent alpine shrubs. The orange dots represent desert shrubs. The blue dots represent secondary shrubs. The red straight line represents the fitting line of alpine shrubs. The orange straight line is the fitting line for desert shrubs. The blue straight line is the fitting line for secondary shrubs (*p* < 0.05). The same applies to the figures below.

**Figure 4 biology-14-01465-f004:**
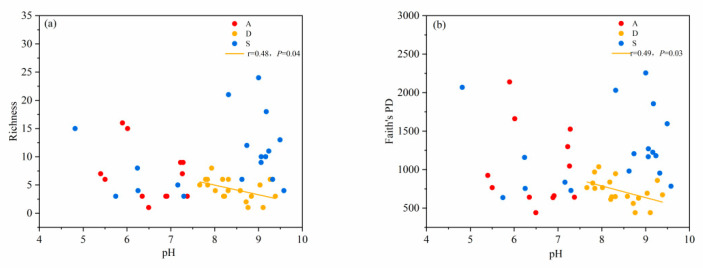
(**a**) Changes in species richness and (**b**) changes in phylogenetic diversity of different shrubland types along pH gradient.

**Figure 5 biology-14-01465-f005:**
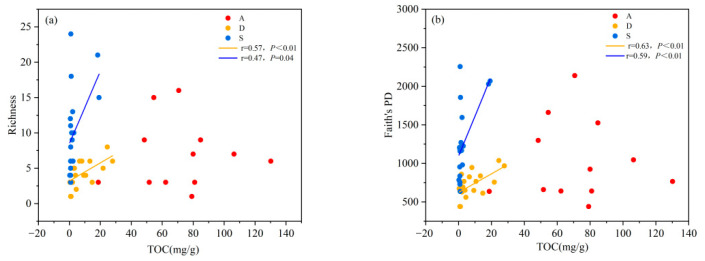
(**a**) Changes in species richness and (**b**) changes in phylogenetic diversity of different shrubland types along the TOC gradient.

**Figure 6 biology-14-01465-f006:**
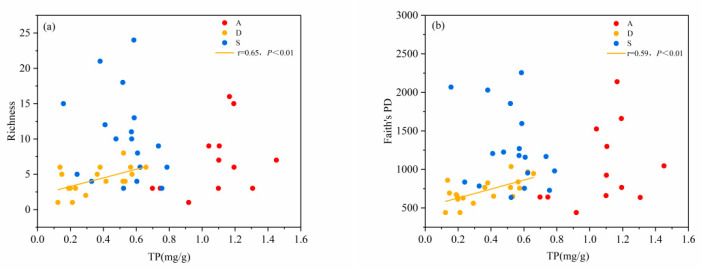
(**a**) Changes in species richness and (**b**) changes in phylogenetic diversity of different shrubland types along the TP gradient.

**Figure 7 biology-14-01465-f007:**
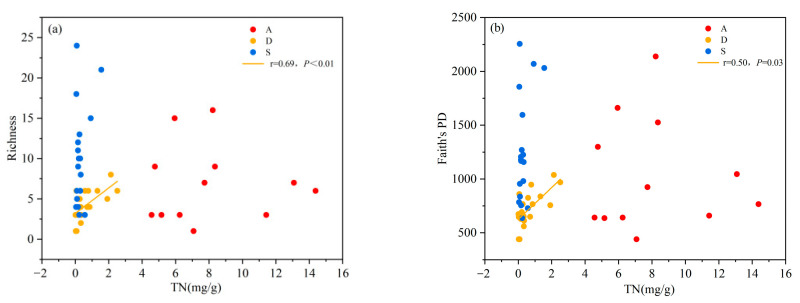
(**a**) Changes in species richness and (**b**) changes in phylogenetic diversity of different shrubland types along the TN gradient.

**Figure 8 biology-14-01465-f008:**
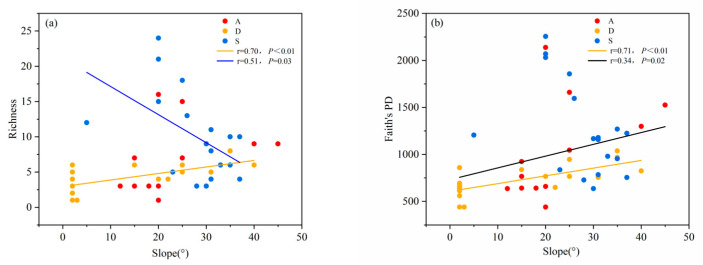
(**a**) Changes in species richness and (**b**) changes in phylogenetic diversity of different shrubland types along the slope gradient.

**Figure 9 biology-14-01465-f009:**
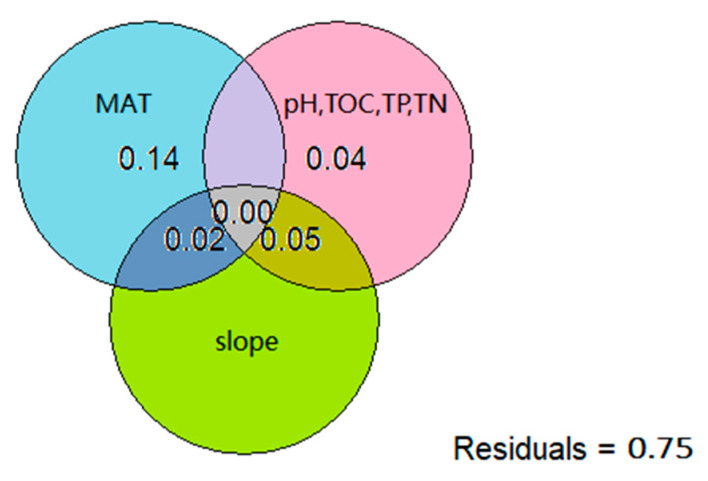
The relative contributions of different environmental factors to shrubland phylogenetic relationships. The blue area represents the climate factor MAT; the pink area represents the soil factors pH, TOC, TP, and TN; and the green area represents the terrain factor slope. The overlapping portions represent the proportions of variance jointly explained by different types of factors. As shown in the figure, 0.14, 0.04, 0.00, 0.02, and 0.05 are the proportions of variance explained.

**Figure 10 biology-14-01465-f010:**
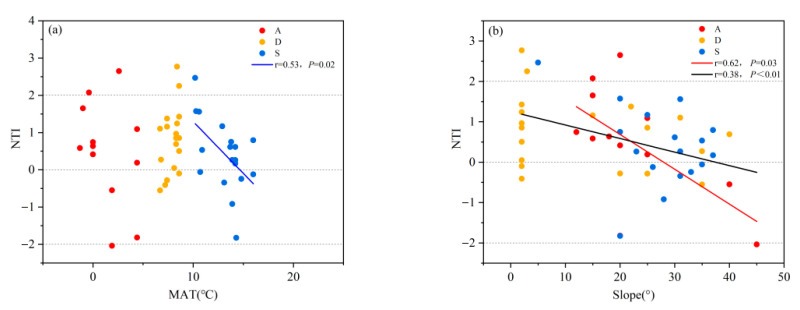
Phylogenetic structures of different shrublands along environmental gradient. (**a**) NTI changes for different shrubland types along the mean annual temperature gradient; (**b**) NTI changes for different shrubland types along the slope gradient.

## Data Availability

Data cannot be provided due to privacy or ethical restrictions. Please contact the corresponding author if needed.

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
