# Peer review of "Drivers of Shrub Community Assembly in Semi-Arid Ecosystems: Integrated Evidence from Environmental Stress on the Western Loess Plateau"

_biology, 2025, doi:10.3390/biology14111465_

Round 1
Reviewer 1 Report
Comments and Suggestions for Authors
Drivers of shrub community assembly in semi-arid ecosystems: 2 Integrated evidence from environmental stress on the western 3 Loess Plateau
Some issues need to be addressed.
The introduction is generally well-written. However, it would benefit from a clearer and more specific set of hypotheses linked directly to the two research questions posed at the end. For example: "We hypothesize that (H1) MAT is the primary driver of diversity across all shrubland types, and (H2) alpine and desert shrublands are primarily assembled by habitat filtering, while secondary shrublands show signs of competitive exclusion due to different disturbance histories."
Methods
In polygenetic tree the method for building the phylogeny is appropriate, but the authors should specify if they resolved polytomies and how.
he null model used for NTI calculations is critical. The manuscript states it was "automatically generated by the software." The authors need to specify which null model was used (taxa.labels, richness, frequency, etc.), as different models test different ecological hypotheses.
The use of simple linear regression for each factor independently is suboptimal, as environmental factors are often correlated (as shown by PCA). A more robust approach would be to use multiple regression or model selection approaches (AIC) to identify the most parsimonious model explaining diversity and NTI.
Sampling Design and Pseudoreplication
The most significant flaw is the treatment of pseudoreplication. The study states that three 5m x 5m quadrats were placed close together (<250m apart) within a single, large (>25m x 25m) shrubland patch, and then the species lists from these three quadrats were pooled into a single "plot" for analysis. This means that each of the 49 "sample sites" is effectively a single, non-independent replicate. The three sub-quadrats are technical replicates of the local community, not independent samples of the shrubland type across the landscape.
This design inflates the degrees of freedom and increases the risk of Type I errors (false positives). The statistical analyses (regressions, variance partitioning) treat these 49 pooled plots as independent data points, which they are not, because environmental conditions and dispersal limitation create spatial autocorrelation. The patterns reported (Figure 3-9) may reflect very local conditions rather than generalizable trends across the western Loess Plateau.
The authors must explicitly acknowledge this as a limitation in the discussion. A more robust design for future work would be to have multiple, truly independent replicate sites for each shrubland type, spaced far enough apart to ensure spatial independence. For the current data, the authors could use the three individual quadrats as samples and employ mixed-effects models with "Site" as a random factor to account for the nested structure. Alternatively, they could use the site-level averages, but this drastically reduces statistical power (from n=49 to n=16-19 per type). The current analysis is valid only if the research question is explicitly about the fine-scale structure within a single patch, which it is not.
Interpretation of Phylogenetic Structure (NTI)
The interpretation of NTI is sometimes oversimplified. The manuscript states that NTI > 0 indicates "environmental filtering" and that this supports "deterministic processes." While habitat filtering is a deterministic process, so is competitive exclusion (which leads to phylogenetic overdispersion, NTI < 0). The finding of phylogenetic clustering (NTI > 0) is consistent with habitat filtering, but it is not the only possible explanation. It could also result from biogeographic dispersal limitation or recent speciation.
The conclusion that deterministic processes dominate is technically correct but too broad. The specific type of deterministic process (filtering vs. competition) is what's interesting.
The discussion should be nuanced. For example, when discussing the shift from clustering to divergence in secondary shrublands with increasing MAT, the authors correctly suggest a weakening of environmental filtering and potentially stronger competition. This is good. This level of interpretation should be applied consistently. The authors should explicitly consider alternative explanations, especially dispersal limitation, particularly for the alpine and desert communities which may be more isolated.
Results
In Figure 2 (PCA), the caption and text state PC1 is related to MAT, pH, slope, and PC2 to MAT, TOC, TP, TN. It's unusual for MAT to load highly on both axes. The authors should check this and clarify the interpretation. The loadings table in Appendix A1 shows MAT has strong negative loadings on both axes (-0.655, -0.779), which is possible but requires careful explanation (it may represent an orthogonal gradient related to MAT's interaction with other factors).
Figure 8 (Variance Partitioning) is confusing. The labels X1, X2, X3 are not defined in the figure itself. The values in the segments (0.14, 0.04, 0.00, 0.02, 0.05) should be explicitly labeled as "Proportion of variance explained."
The discussion is generally good but could be tightened. The speculation about mechanisms (evaporation in deserts, human disturbance in secondary shrublands) is appropriate but should be clearly framed as plausible hypotheses rather than proven facts.
The contradiction between this study's slope findings (greater slope -> more divergence) and other studies (greater slope -> more clustering) is interesting. The authors' explanation about microhabitat heterogeneity and reduced human disturbance on steep slopes is excellent and should be highlighted as a key finding.
Reviewer 2 Report
Comments and Suggestions for Authors
Dear authors,
Thank you for submitting your manuscript entitled “Drivers of shrub community assembly in semi-arid ecosystems: Integrated evidence from environmental stress on the western Loess Plateau.” I found your study to be timely, relevant, and well-structured, addressing an important ecological question regarding shrubland diversity and community assembly under environmental stress. The integration of species diversity, phylogenetic structure, and environmental drivers is a clear strength of your work, and the findings contribute valuable insights to the field.
That said, while the manuscript is promising, several sections would benefit from further refinement to enhance clarity, depth, and scientific impact. My detailed comments regarding strengths, limitations, and suggested improvements have been provided in the attached review file for your consideration. These cover points across the abstract, methods, discussion, and conclusions, including recommendations for improving the flow, emphasizing novelty, clarifying methodological justifications, and highlighting practical implications.
Overall, I believe this is a solid manuscript that, with revision, has strong potential for publication. I encourage you to carefully address the comments provided to improve the readability, rigor, and broader significance of your work.
• Summary of the Manuscript
The paper looks into the variety of shrublands and how communities form in the
western Loess Plateau when the environment is stressed. It does this by combining
species and evolutionary diversity from desert, alpine, and secondary shrublands. It
gives us useful ecological information from an area that is not studied very much. It
also shows how deterministic processes and habitat filtering are connected to
community structure, which has implications for management and restoration.
• Comments on all text sections:
Abstract:
Strong ecological context, but too convoluted language. Key concepts are not clearly
defined, the research gap is unclear, and methodological context is limited.
Introduction:
Excellent background and reason for merging diversity metrics. However, it is
lengthy, repetitious, and does not explain how this study is innovative.
Methods:
1-The criteria for site selection, such as "typical vegetation" and "mild
disturbance," are subjective and insufficiently explained.
2-Pooling and quadrat size have the potential to overlook regional variation.
3-The soil samples are composited, which tends to hide fine-scale variation;
climatic data are solely dependent on MAT, which has its limitations.
4-Linear models and a single phylogenetic index are the primary components of
statistical analysis, with relatively little justification for null models.
Discussion:
Results are contextualized within ecological theory; however, they are frequently
repeated in a descriptive manner. Certain interpretations are speculative, and the
model's assumptions, climate variables, and sample size are not addressed.
Conclusion:
Mostly just repeats the results without pointing out anything new, any problems, or
how they could be used in management.
Constructive Feedback for the Authors
Abstract clarity: Shorten sentences, define technical terms, highlight the research
gap and novelty, and briefly describe the methods.
Introduction focus: Reduce repetition, tighten the narrative, and clearly articulate
how this study advances understanding beyond prior work.
Methods transparency:
• Explain why you chose the spot and the size of the quadrat.
• Explain how disturbances were measured and managed.
• Establish objective criteria for the selection of "typical" vegetation.
• Detail soil/laboratory protocols and expand climate variables (e.g.,
precipitation, aridity indices).
Statistical analysis:
• Explain the choice of the null model and the GLM parameters.
• Nonlinear models, such as GAMs, can be used to test stability.
• To partition variance, use several phylogenetic and diversity measures.
• Perform diagnostic tests, such as residual plots.
Discussion improvements:
• The comparison should be critical with previous studies, with an emphasis
on unique discoveries.
• It is important to state restrictions and uncertainty explicitly.
• Avoid making interpretations that are hypothetical and are not explicitly
supported by data.
Conclusion:
In this section, you will highlight the distinctive contributions that the study has
made, explicitly describe the limitations of the study, and provide implications for
the management and conservation of shrubland.
Reviewer 3 Report
Comments and Suggestions for Authors
Your manuscript addresses community assembly processes in shrublands of the Loess Plateau, an ecologically important region. The study is timely and relevant, and the comparative approach across different shrubland types is a strength. However, in its current form the paper requires substantial revision before it can be considered for publication.
The introduction provides some useful context, but it is overly descriptive and repetitive. It should be streamlined to highlight the specific research gap and to justify more clearly why this study is needed. Some relevant recent and regional studies are missing from the references, particularly in the Chinese ecological literature, which would strengthen the contextualisation.
The methodology section is insufficiently detailed. The criteria for site and quadrat selection, the relatively small sample size, and the rationale for choosing the environmental factors analysed are not adequately explained. Without this information, it is difficult to assess the robustness of the design. In addition, your statement that data cannot be shared for “privacy or ethical reasons” is problematic in ecological fieldwork. Data availability is fundamental for reproducibility; you should provide at least anonymised or partial datasets or a clear and convincing justification.
The results are generally presented in an organised way, with figures and tables supporting the analysis, but some redundancy remains. Captions should be expanded so that figures are understandable without reference to the main text. The conclusions are mostly consistent with the results, but at times they are too broad, claiming stronger determinism than the evidence supports. Greater caution and more precise wording are needed.
A significant weakness is the language. The manuscript is readable but hampered by awkward phrasing, redundancy, and grammatical errors. We strongly recommend thorough revision of the English by a fluent speaker or professional editor to improve clarity and flow.
In summary, your study has merit and could contribute to the ecological literature on shrubland assembly in the Loess Plateau. However, improvements are needed in the clarity of the introduction, methodological transparency, strength of the analysis, use of local references, and overall language quality. Addressing these issues will considerably enhance the paper and make it suitable for publication.
Comments on the Quality of English Languagenone
